# Sustaining International Students' Learning of Chinese in China: Shifting Motivations among New Zealand Students during Study Abroad

**Yang (Frank) Gong [1] , Mengyao Ma [2,\*], Tien Ping Hsiang [1,\*] and Chuang Wang [1]**

[1] Faculty of Education, University of Macao, Macao SAR, China; frankgong@um.edu.mo (Y.F.G.); wangc@um.edu.mo (C.W.)

[2] School of Foreign Languages and Literature, Wuhan University, Wuhan 430072, China

\* Correspondence: 2015101020010@whu.edu.cn (M.M.); tphsiang@um.edu.mo (T.P.H.)

**Abstract:** This paper reports on an inquiry that examined groups of New Zealand students' motivational shifts related to learning Chinese before and after relocation to China. In the inquiry, we encouraged 15 participants to write reflective journals and conducted two rounds of interviews before and after their study abroad trip to China. The analysis revealed that most participants had their motivation enhanced by the trip, and expected to sustain their heightened motivation for learning Chinese in the future. The findings suggest that the participants' motivational shifts happened during their period of study abroad in China, and were prompted by their new pedagogical environment and individual learning experiences. In other words, the motivational enhancement emerged from ongoing interactions between the participants' L2 self-concepts (e.g., ideal L2 selves) and learning and sociocultural contexts in China. These findings offer fresh insights into the dynamic nature of Chinese language learning motivation and the role of formal and informal settings in the participants' learning of Chinese. They imply that educational stakeholders need to provide authentic communication opportunities and resources to enhance international students' motivation for the sustainable learning of Chinese.

**Keywords:** learning motivation; motivational fluctuations; Chinese as an additional language; study abroad

## 1. Introduction

Over the last two decades, Chinese has emerged as an increasingly important language taught and learnt as an additional language around the world [1–3]. By the end of 2018, more than 2.7 million people from 154 countries/regions were reportedly learning Chinese as an additional language (CAL) through various learning modes [4]. Despite the rising number of CAL learners both in and outside China, CAL teachers have reported a number of challenges that are related to this study, such as difficulties in generating and sustaining learning motivation [5], low learner retention rates [6–8], and a lack of research on Chinese language learning motivation [9,10]. For this reason, there is a pressing need for researchers to explore CAL learners' motivation in diverse teaching and learning contexts [5].

Previous research has documented the motivational shifts of second/foreign language (SL/FL) learners when they travel to learn languages in new contexts, such as the host contexts for short-term study abroad students (e.g., [11,12]) and long-stay students pursuing academic degrees (e.g., [13,14]). These studies have mainly examined the motivational trajectories of students who move from their home environment to an English-speaking context, or from English-speaking countries to a European-language speaking country like France (e.g., [11]) or Spain (e.g., [15]). However, little attention has been paid to the motivation of CAL leaners who relocate to China for immersion

experiences or academic degree studies [9]. Moreover, studies on motivation in various contexts have concentrated on investigating what motivation is, how to measure learning motivation, and the relationship between learner factors (e.g., gender, age, sociocultural backgrounds, strategy use, learning achievement) and motivation [8,16]. Relatively few studies have focused on the interaction between learner motivation and contextual conditions, especially the study abroad context.

Previous research on CAL learners' motivation has been guided by a traditional social-psychological perspective and has tended to define motivation in a comparatively static fashion, and hence, it has been "unable to capture the fluctuation and multifaceted complexity of (the) Chinese learning motivational process" [8] (p. 365). At the same time, most of the extant knowledge about CAL learners' motivation thus far is based on research about learners' experiences in non-Chinese speaking settings, such as the United States and Europe. International students in China remain one of the most underexplored populations to date [10].

To address the gaps described above, the present study investigated the Chinese learning experiences of a group of New Zealand students who were studying abroad in a university in China, and examined the dynamic nature of their motivation to learn Chinese during their academic stay in the country. Specifically, this research aimed to illustrate the evolving Chinese learning motivation of study abroad learners, and its interaction with the various contextual conditions surrounding them in China. It can further our understanding of CAL learning motivation in particular and SL/FL motivation in general, and thus provide practical applications for CAL teaching and learning worldwide.

## 2. Literature Review

### 2.1. Motivation in Second/Foreign Language Education

Motivation is widely seen as a critical component in successful second/foreign language (SL/FL) learning [17–22]. Early understandings of SL/FL learning motivation have been largely influenced by a social-psychological perspective, especially Gardner and Lambert's [23] conceptualizations of integrative and instrumental motivation. Integratively motivated learners have positive attitudes toward the target language community and learn the language to communicate and identify with members of that community, whereas instrumentally motivated learners learn a language for potential pragmatic benefits, such as having a better job or a higher salary.

Motivation has also been theorized to be situated on a continuum of intrinsic motivation, extrinsic motivation, and amotivation, which "vary in the extent to which the goal is controlled by the individual or by external contingencies" [24] (p. 425; also see [25]). According to Deci and Ryan [25], intrinsic motivation is learning a language for the inherent enjoyment and personal satisfaction of doing so. Extrinsic motivation has three sub-categories, including external regulation, introjected regulation, and identified regulation. While external regulation means the learning is determined by achieving a reward, introjected regulation refers to performing an activity due to internal pressure, such as guilt or self-enhancement. Identified regulation means that individual learners carry out an activity for personally meaningful reasons. For example, they want to learn another language, or they believe that becoming bilingual/multilingual is a valued goal of personal development. According to Deci and Ryan [25], identified regulation is the most self-determined category of extrinsic motivation, while amotivation describes a lack of intrinsic or extrinsic motivation in the learning process.

These conventional views have generally been criticized for being too simplistic to capture the variability and complexity of SL/FL learning motivation with an emphasis on learners [8,14,22,26]. Many researchers have stressed the importance of understanding one's learning motivation as a "socially constructed, dynamic" process [14] (p. 600; also see [27]). Drawing upon Higgin's [28] self-discrepancy theory and other major motivation theories (e.g., [29]), Dörnyei [30] expanded the realm of SL/FL learning motivation research and proposed the L2 Motivation Self System (LMSS). To capture the dynamic and multifaceted nature of L2 motivation, the LMSS consists of three primary components of motivation: ideal L2 self, ought-to L2 self, and L2 learning experience. The ideal L2

self refers to the L2-specific aspects of one's ideal self. It can be represented by an integrative or instrumental motivation with a promotion focus, denoting a learner's aspirations, hopes, and wishes to achieve growth or success. The ought-to L2 self concerns "the attributes that one believes one ought to possess in order to avoid possible negative outcomes" [31] (p. 29). While the ideal L2 self and the ought-to L2 self are considered to be motivational constructs, L2 learning experience is concerned with "situation-specific motives related to the immediate learning environment and experience" [31] (p. 29), such as social learning environment or classroom settings. Given its emphasis on the dynamic nature of SL/FL learners' motivation and the role of interactions between motivation and learning contextual conditions, the LMSS is particularly relevant to the context of the present research, where the participants moved to the target language community and gained access to members of the community (e.g., [10,13,27]).

### 2.2. Motivation for Learning Chinese as an Additional Language

Only a few of studies on Chinese as an additional language (CAL) learners' motivation have been conducted since the 1990s. Early research generally applied a social-psychological perspective and focused on the relationship between CAL learners' individual factors and their learning motivation, or on the impact of motivation on their Chinese linguistic competence [8]. For instance, Wen [7] explored the different patterns of intrinsic motivation between beginning and intermediate CAL students in a university in the United States, and found that the students' intrinsic interest in Chinese culture and their desire to understand their own cultural heritage were connected to their motivation to start learning Chinese, but not to the motivation to continue learning Chinese. Yu and Watkins [32] focused on international students studying abroad in China and indicated the significant association between their integrative motivation to learn about Chinese culture and their Chinese language proficiency.

In recent years, CAL researchers have expanded their view to examine motivation fluctuations based on learning experience, and have paid attention to the role of instructional settings such as educational subject dimensions. Cai and Zhu [9] employed the LMSS as a framework to explore the impact of an online learning project on university students' CAL learning motivation in the United States, and reported that L2 learning experiences changed significantly before and after the online project. Ruan et al. [5] examined the influence of task-based instruction on beginner CAL learners' motivation in a Danish university. The result demonstrated that unfocused tasks and tasks involving group work, interactions, learner autonomy, and cultural elements could motivate the learners. Building on these existing studies of CAL learners' motivation, more studies are needed to explore interactions between motivation, learning experience, and the learning context [8,9].

Existing studies on CAL motivation have not employed a socio-dynamic perspective, and have thus failed to offer a realistic account of motivational phenomena and interactions with the target language community [10,33]. This important gap invited further inquiry in relation to shifts in CAL learners' motivation and the role of contextual conditions in shaping motivation in the study abroad context. To address these gaps, the present research addresses the following two questions:

- RQ1: How does the participants' motivation for Chinese learning change during study abroad in China?
- RQ2: How does the participants' motivation for Chinese learning interact with their learning and sociocultural contexts during study abroad in China?

## 3. Methodology

### 3.1. Research Context and Participants

The current study focused on the motivational shifts of a group of New Zealand students who moved from an English-speaking context to a university in China to learn Chinese. These students were from a Chinese as an additional language program offered by a university in New Zealand. This program offered Chinese language courses ranging from beginner to advanced level, through either

face-to-face or online distance learning mode, and with students from each mode of course delivery using the same learning materials and studying at the same pace. With China's rapid economic growth and increasing international influence in the world, especially in the Asia-Pacific area, more and more New Zealand students are keen to enhance their Chinese language proficiency and cultural knowledge through study abroad programs in China. To fulfill New Zealanders' needs for Chinese language and culture learning, the New Zealand government has offered the Prime Minister's Scholarship for Asia (PMSA) since 2013, awarded to individuals learning Chinese or conducting Chinese studies in China.

With the aim of promoting their Chinese proficiency and learning about Chinese culture, our 15 participant students had all obtained PMSA support and were voluntarily participating in a study abroad program in a university in China, which lasted around six weeks. The study abroad program mainly focused on enhancing the students' listening, reading comprehension, and Chinese character writing ability, and all courses were taught by native Chinese teachers with Chinese as the only medium of instruction. At the same time, videos, music, literature, and field trips to historic sites were used to facilitate their historical, social, and cultural understanding of China. The participants were selected according to the following criteria: (1) they were native English speakers studying Chinese in China; and (2) they agreed to share their opinions and experiences regarding their Chinese learning motivation change before and after relocation to China, and their motivational fluctuations during their study abroad period. Consent forms were obtained beforehand, and all the participants were assured of the confidentiality and anonymity of the research.

Since the 15 participants had limited exposure to authentic Chinese-speaking settings, Chinese community, or Chinese culture prior to their relocation in China, they had little knowledge about the sociocultural structure in China, such as daily living, communication styles, and the educational system. For example, from their self-reports we noted that some of them had preconceived ideas that English is widely used in China, especially in developed cities like Beijing and Shanghai.

Details of the participants can be found in Table 1, showing that the participants were heterogeneous regarding personal background demographics, such as their age, Chinese proficiency level, and major.

**Table 1.** Participants' profiles.

| No. | Name | Age | Gender | Study Mode | Language Level | Major |
|---|---|---|---|---|---|---|
| 1 | Ella | 18 | Female | Internal class | Beginner | Arts |
| 2 | Ember | 59 | Female | Internal class | Lower intermediate | Linguistics |
| 3 | Zoey | 22 | Female | Distance class | Intermediate | Arts |
| 4 | Isla | 23 | Female | Distance class | Intermediate | Chinese and Japanese |
| 5 | Molly | 24 | Female | Distance class | Beginner | Arts |
| 6 | Clara | 44 | Female | Distance class | Intermediate | Arts |
| 7 | Stella | 19 | Female | N/A | Lower intermediate | Design |
| 8 | Bennett | 20 | Male | Internal class | Beginner | Information Science |
| 9 | Hunter | 28 | Male | Internal class | Beginner | Arts |
| 10 | Parker | 31 | Male | Internal class | Intermediate | Business |
| 11 | Austin | 29 | Male | Distance class | Intermediate | Arts |
| 12 | James | 44 | Male | Distance class | Advanced | Arts |
| 13 | Ashton | 45 | Male | Distance class | Advanced | Chinese |
| 14 | Wesley | 51 | Male | Distance class | Beginner | Arts |
| 15 | Jackson | 54 | Male | Distance class | Intermediate | Arts |

Note: All names are pseudonyms.

### 3.2. Data Collection

Concentrating on the New Zealand students' motivational trajectories related to Chinese learning prior to and after their learning context change, this research sought to elaborate on and interpret their experiential accounts in terms of their Chinese learning, from planning to taking part in the study abroad program until after the six-week academic sojourn; this took place immediately after they were accepted into the program in New Zealand. The study consisted of three phases of data collection in

order to obtain a "thick description" and a holistic and deep understanding of the phenomenon under research [34]. In the first stage, the first author interviewed all the participants about their Chinese learning and use experiences in New Zealand. During the interviews, the following topics were addressed: (1) motivation to learn Chinese and improve Chinese proficiency; (2) attitudes to Chinese language and Chinese community; (3) Chinese learning and use experiences; (4) current academic achievement/development; (5) reasons for studying Chinese in China; and (6) expected outcomes of study abroad.

In stage two, the participants were encouraged to write reflective journals during their stay in China. A reflective journal is both a product and a process, which helps researchers "capture an experience, record an event, explore our feelings, or make sense of what we know" [35] (p. 9). In particular, it encouraged our research participants to elaborate and document their motivational shift and fluctuations prior to and after relocating to China [13,14], to consider and analyze their L2 selves and their interaction with their Chinese learning experiences and the learning context [31], and to reflect on and articulate their future plans regarding Chinese learning and use. These advantages are consistent with the two research questions posed by the study. To capture the participants' evolving learning motivation and their interaction with the learning environment in China, four questions/prompts were used to guide their reflections: (1) What are your Chinese learning and use experiences in China? (2) What is your learning achievement in China? (3) How do you describe your learning motivation change in China? (4) In your mind, what is the most challenging to adjust to in the learning environment of China? The reflective journals were collected each week and comprised 86 entries in total. Each entry was about one single-spaced typewritten page in length, and each was identified with a code—for example, "Stella/R/W6" meant the entry was Stella's reflection entry from the sixth week of her academic sojourn.

In the third phase, the first researcher interviewed all the participants within a week after the study abroad program ended, to minimize possible memory biases. The interviews focused on the differences in their Chinese learning motivation before and after their academic sojourn in China. The following questions were asked in these interviews: (1) Were your Chinese learning and use experiences in China similar to your previous experiences in New Zealand? (2) Is your current Chinese learning motivation similar to your previous motivation before you studied abroad? (3) Are your Chinese learning efforts similar to your previous efforts in New Zealand? (4) Is your current Chinese learning plan similar to your previous plan in New Zealand?

All the interviews conducted in stages one and three were carried out in the participants' native language, English, to minimize language barriers. These interviews were audio-taped, transcribed verbatim into English, and double-checked for accuracy. Participant-checking procedures were undertaken after we had transcribed the interviews, to enhance the rigor of the research and the trustworthiness of the subsequent analysis [36].

### 3.3. Data Analysis

NVivo 12 was used to analyze the data. To examine and track changes in the participants' Chinese learning motivation and the interaction between various motivational components and the learning context, this research followed an open-coding approach [13,37]. In other words, the coding and recoding processes were not restricted to any predetermined categories.

All the journal entries and interview transcriptions were read through five times, and the coding process was informed by the research questions and the literature on the participants' evolving Chinese learning motivation and its association with the learning setting. The coding process began by reducing the data to various descriptive categories, such as "motivation to learn Chinese in New Zealand and China" and "future plans". Then, the relationships between different initial nodes were identified and higher-order themes were generated from the lower-order nodes [38]. For instance, "a temporary frustration in learning Chinese" was categorized under a higher-order node, "motivational fluctuations

during academic sojourn in China"; and "becoming a fluent Chinese speaker" and "becoming a competent communicator with locals" were clustered under "ideal L2 self".

When different categories of Chinese learning motivational changes and elements had been identified, the interaction patterns of these motivational components were analyzed. At the same time, the links between "ideal L2 self" and relevant contextual nodes such as "academic setting" were analyzed, and these nodes were arranged under another higher-order node, "interplay between ideal L2 self and context". The existing nodes were constantly revised, deleted, or merged with others as new categories emerged. The coding and analysis were compared across the participants and repeated until saturation was reached. Throughout the data analysis, annotations and memos were used to record immediate comments and reflexive thinking on the data, and they were further used to assist data coding and categorization [39].

## 4. Findings

Overall, the data analysis indicated that during the participants' academic sojourn, their Chinese learning motivation showed a significant enhancement after arrival and various fluctuations during their stay in China. Twelve of them (12/15) reported a heightened motivational intensity after study abroad, and stated that they expected to sustain their boosted level of motivation for their future Chinese learning after returning to New Zealand. From time to time, they also experienced motivational surges caused by contextual conditions and personal learning experiences, such as new pedagogical environments and students' interactions with teachers in the classroom. It should be noted that the participants usually returned to their initial motivational intensity when perceiving the improvements they had made in their Chinese learning during the academic sojourn.

The participants' accounts suggested that their ongoing motivational changes and surges were elicited and shaped by the continuous interaction between their ideal L2 selves, their L2 self-concepts, and the study abroad context. After arrival in China, most of them (12/15) displayed an instrumental-value direction in terms of their Chinese learning motivation, albeit with different learning purposes. Almost half of the participants (7/15) envisaged more elaborate ideal L2 selves related to high Chinese proficiency (e.g., being a fluent Chinese speaker) during their stay, focusing on listening and speaking skills. In order to reduce the existent discrepancy between their L2 self-concepts and their ideal L2 selves, they made various efforts to seek opportunities and resources in and outside the classroom to promote their oral Chinese. Their continuous motivational shift occurred as a result of their interactions with their learning and sociocultural contexts.

### 4.1. Motivational Change after Arrival in China

#### 4.1.1. Motivational Enhancement from New Zealand to China

Comparing the participants' Chinese learning motivation prior to and after their academic sojourn, it was found that most of them (12/15) reported that their motivation was enhanced after relocating to China, and they also expected to maintain their heightened motivation after returning to New Zealand (see Table 2). In particular, after their arrival in China, except for one participant who did not report a motivational change, all the other participants reported an increase in their motivational intensity towards Chinese learning. A few weeks later, they noted in the last week of their stay:

1. One thing is for sure: my motivation to study Chinese is stronger than ever!. (Isla/R/W6)
2. As a wise man once said and which I will appropriate for my journey into Chinese: "This is not the end, this is not even the beginning of the end, but it is perhaps the end of the beginning!". (Austin/R/W6)

**Table 2.** Shifting motivation before and after relocation in China.

| Name | New Zealand | China |
|---|---|---|
| 1. Ella<br>Working in a New Zealand coffee machine company with business connections with China | Only focused on the Chinese language itself | Increasingly interested in Chinese language and its culture |
| 2. Ember<br>Cantonese heritage speaker, staff member at a university in New Zealand | Learnt Chinese to communicate with international students from China | Growing interest in speaking Chinese |
| 3. Zoey<br>Working in an agriculture company with business connections with China | Understood Chinese vocabulary and grammar points | Strong desire to become a fluent Chinese speaker |
| 4. Isla<br>Student majoring in Chinese and Japanese | Interested in learning different languages | Expected to use Chinese in her future work |
| 5. Molly<br>Working in a technical company with business connections with China | Used Chinese to communicate with cooperative partners in China | Learnt Chinese language and culture |
| 6. Clara<br>High school teacher teaching Japanese, farm tourism business owner | Used Chinese to communicate with tourists from China | More determined to enhance listening and communicating skills |
| 7. Stella<br>Heritage Chinese student | Learnt Chinese to communicate with Chinese relatives | Growing learning desire, Chinese learning as a means to enhance heritage identity |
| 8. Bennett<br>Heritage Chinese student | Interested in Chinese culture and people | Increasing desire to enhance Chinese proficiency through immersion |
| 9. Hunter<br>Beginner Chinese learner | Interested in learning about Chinese people | Growing interest in learning Chinese language and culture, and expected to move to China |
| 10. Parker<br>Had a Chinese wife, planning do business with Chinese people | Expected to achieve high level of Chinese proficiency | Increasingly interested in Chinese culture and lifestyle and aimed to be a fluent Chinese speaker |
| 11. Austin<br>Learner of multiple languages | Challenged himself through Chinese learning | Aimed to be an effective communicator |
| 12. James<br>Had a Chinese wife | Learnt Chinese to communicate with Chinese family members and people | Listening, speaking, reading and writing skills in Chinese |
| 13. Ashton<br>Cricket coach | Strong interest in Chinese language | |
| 14. Wesley<br>Lawyer, mature learner | Interested in learning Chinese language | Used Chinese to further his career |
| 15. Jackson<br>CEO of a small company, doing business with Chinese companies | Understood Chinese culture and business etiquette | Strong desire to enhance Chinese ability to engage with Chinese trading partners |

While both Isla and Austin had showed an interest in learning different languages prior to studying abroad, their six-week academic trip from New Zealand to China boosted their Chinese learning motivation. Their Chinese language learning and use experiences in authentic settings offered them more participation and strengthened their tangible connections with the Chinese community.

Some participants, like Clara and Hunter, also contrasted their motivational shift before and after relocating to China, describing the study abroad experience as "fuel" for learning Chinese:

3. Before this experience, I was a little on the fence on the future of my Chinese language studies, however, this discovery (that my Chinese language learning journey is still continuing) has only added fuel to the flame for me and made me realize I have a lot more I want to learn. I am more determined than ever to continue with my studies with the focus being on improving my listening and communication skills. (Clara/R/W6)

Here, Clara is expressing her desire to continue investing in Chinese language learning. Hunter even "planned to move to China in the future" to experience "the culture first-hand" (Hunter/R/W6). In this sense, his learning purpose had expanded from Chinese proficiency alone to integrating himself into the host community.

The data also revealed that the participants' motivational intensity was closely related to their Chinese learning goals, and motivational enhancement occurred as the learning goals became more tangible. For instance, Ella was an employee of a coffee machine company that had business connections with China, and she had solely concentrated on her language self before studying in China. However, at the end of her stay, she expressed a strong motivation towards understanding Chinese culture, and summarized her learning goals beyond the language-only viewpoint in the last week of her study abroad period:

4.　This experience in its entirety has allowed me to gain a cognizance and appreciation for Chinese culture. . . . One strategy that I am going to implement is to read the Chinese news on a weekly basis. Through this way, I will be able to get regular exposure to Chinese language and culture/foreign affairs. I will be building up my cultural intelligence as well as developing and maintaining my language skills. (Ella/R/W6)

The study abroad experience had expanded the scope of her Chinese learning and made her learning purposes more specific and explicit, such as having "a deeper appreciation and knowledge of Chinese food and etiquette" (Ella/R/W2) and learning "common classroom terminology" (Ella/R/W3). Hence, her learning motivation was increased in terms not only of linguistic knowledge, but also of intercultural understanding.

In a similar vein, Parker had a Chinese wife, and during his initial interview in New Zealand, he reported an extrinsic motivation to learn Chinese in order to do business with Chinese people. During his stay in China, he realized the importance of the social and cultural aspects of Chinese learning, such as, "understanding of Chinese culture and lifestyle" (Parker/R/W6), and expected to sustain his enhanced motivation by setting explicit learning goals after going back to New Zealand:

5.　When I return back to New Zealand, I know I won't be surrounded by the Chinese language, so I have downloaded a lot of Chinese music so I can still practice my listening skills and singing in Chinese. (Parker/R/W6)

It also should be noted that other participants (Ember, Clara, Stella, and James) who had general intentions to improve their communication ability before study abroad reported more specific goals, such as improving their listening or speaking skills.

4.1.2. Motivational Fluctuations During Study Abroad in China

Even though the participants' Chinese learning motivation generally increased significantly after moving to China, many of them reported various motivational fluctuations during their six-week stay because of contextual conditions or specific learning experiences in the host community.

There were a range of differences between the New Zealand and Chinese educational environments. In Chinese classrooms, the teacher is normally seen as a model and authority [40]. Specifically, in the Chinese as an additional language classroom, rote learning, repetition, and memorization are usually adopted by Chinese teachers. Ashton was the only participant who was majoring in Chinese in this program and he demonstrated a strong interest in learning Chinese, but he experienced a motivational decrease after taking part in a self-introduction activity in the Chinese speaking class:

6.　*She (the teacher) kept saying it louder and louder, getting me to repeat it, thinking by getting louder and slower I would say her version. By this stage, I was very annoyed and so kept repeating the wrong sentence, and in the end, I just said, "I forgot it" and looked away. I could not say, "I am trying to say this" in Chinese. . . . The result was I was annoyed, and it has made the class less enjoyable for me as I go there to learn, not to be put on the spot in front of the group. I am aware we need to speak more and more Mandarin, but I have never enjoyed public speaking and have turned down speaking engagements. (Ashton/R/W3)*

As noted, combined with the teacher's possible lack of knowledge about their students' linguistic proficiency level, this teacher-centered instructional style reduced Ashton's willingness to participate in the classroom learning practices. Consequently, his Chinese learning motivation decreased due to the "annoying" teacher-student interaction, although he recognized the significance of practicing spoken Chinese with the teacher's assistance in the classroom. The influence of this motivational decrease did not seem to disappear throughout his academic sojourn, because Ashton did not mention any motivational recovery or enhancement regarding Chinese learning in his summary (see Table 2).

Moreover, since the course workload was often beyond their usual learning pace in New Zealand, participants' motivational decreases usually emerged in the early stage of their study abroad period,

and disappeared immediately after they became adapted to the new environment. Zoey was an intermediate Chinese learner, and she expressed her frustration with learning Chinese during the second week of her stay:

7.　　The volume of new words and grammar points being learned at once is a lot larger than what we are all used to. ... In one instance last week, I found myself unable to gain the meaning of the Chinese text while we were reading it aloud due to the large amount of new vocab and new sentence structures used. This frustrated me. (Zoey/R/W3)

However, she later recognized her achievement in "slowly overcoming my fear of using incorrect grammar" in language use (Zoey/R/W6), and she reported her desire to "join a Chinese professional group" to continue improving her Chinese proficiency after returning to New Zealand. Her sense of achievement played a positive role in helping her to overcome the challenges in the Chinese pedagogical environment which had initially reduced her learning motivation.

In short, most of the participants reported that their Chinese learning was motivated by a desire to improve their Chinese language, but they often experienced motivational decreases when facing academic challenges during study abroad.

*4.2. Interplay of Chinese Learning Motivation, Self, and Context during Study Abroad in China*

The analysis indicated that all the participants' Chinese learning motivation became increasingly diverse, but most of them (12/15) displayed an instrumental orientation to learn Chinese after arrival in China. While three participants (Molly, Ella, and Parker) claimed their initial focus was on Chinese linguistic content in New Zealand, all of them agreed that they became more interested in Chinese culture after relocating to a Chinese community. Additionally, Isla, Hunter, and Wesley reported that their Chinese learning motivation was increasingly connected to the learning and social contexts of China. Before studying in China, they had all reported an intrinsic motivation to learn Chinese (e.g., an interest in learning languages), but they showed a strong career-related learning motivation during the academic sojourn.

For instance, Isla was following a double major in Chinese and Japanese and was interested in different languages. As one part of the Chinese culture study, Isla and her classmates visited a diary company in China, and this experience made her "truly recognize the efficiency and high standards that China has" (Isla/R/W5). She perceived that the visit was an important cultural learning experience, which provided her with an accessible opportunity to consider "China-related jobs":

8.　　To link this broad realization back to my own learning and goals, I think I should consider more carefully what my goals are. The main one is that I definitely want to be able to use Chinese in future work, even if it isn't in a huge factory (most likely not!). (Isla/R/W5)

The study abroad experience helped Isla to appreciate, as a native English speaker, that being fluent in the Chinese language or a bilingual communicator represents cultural capital in China [41,42], which could benefit her career development. As her current and future association with the target language community became more practical and close, the tangibility of Isla's cultural capital increased. Accordingly, she appreciated that her learning efforts were more meaningful and her Chinese learning motivation became "stronger than ever!" (Isla/R/W6). For other participants (5/15), their extrinsic motivation was enhanced by their stay in the host community and they reported that Chinese language proficiency represented financial, social, and symbolic capital to connect New Zealand to China.

According to Dörnyei [31], language learning motivation represents a learner's desire to bridge the discrepancy between the ideal L2 self and the L2 self-concept, and therefore, it is also crucial to understand an individual learner's L2 self-concept. L2 self-concept is regarded as "an individual's self-descriptions of competence and evaluative feelings about themselves" in relation to the learning of an L2 [43] (p. 14). Before moving to China, Isla, Bennett, Hunter, Austin, Ashton, and Wesley (6/15) all reported their intrinsic motivation to learn Chinese (their interest in the language); Ember, Clara, Stella,

James, and Parker (5/15) had a communication-orientation learning motivation; Zoey and Ella (2/15) focused on Chinese linguistic knowledge; and Molly and Jackson (2/15) showed a strong extrinsic learning motivation related to their future careers. Although the participants showed different profiles in terms of their Chinese learning motivation, most of them had not formed elaborate ideal L2 selves in New Zealand. They generally had little awareness of their L2 self-concepts given that they were living in an English-speaking context, and some participants even felt frustrated after finding out that Chinese was the sole dominant language in the host community.

9. I was at somewhat of an understanding that English would be commonly spoken. However, I soon came to realize that I had very little, if any, understanding at times, which proved to be frustrating, and in most cases, the other party could not speak English either. (Ella/R/W6)

However, the challenges concerning language learning and usage that the participants encountered during study abroad caused them to discover that their Chinese skills were insufficient, especially at the beginning of their stay:

10. My first experience that really leads me to struggle while being here in China was communicating my ideas across to the local people in Chinese. This was a huge problem because I needed to order food and a beverage at the canteen with what little Chinese I knew. (Bennett/R/W1)

11. I remember the first lesson, I was extremely overwhelmed with all the Chinese speaking. You have to be paying full attention to what the teacher is saying at all times. As soon as the first lesson was finished I remember saying to the others, "Wow, my brain is fried". (Stella/R/W3)

Experiencing a range of negative feelings in the target language community, almost all participants considered their initial Chinese language learning and use in New Zealand to be unsuccessful and unsatisfying. This experience stimulated them to construct more elaborate ideal L2 selves and find more effective strategies to enhance their Chinese proficiency when they became more aware of the dominant role played by the Chinese language in Chinese academic and social settings.

Almost half of the participants (7/15) imagined an idealized self-image with a good command of Chinese. For instance, Parker expected to "achieve the goal of becoming a fluent Chinese speaker" (Parker/R/W6), Zoey defined her academic sojourn as "a journey to Chinese fluency" (Zoey/R/W6), and Clara's learning objective was to become a competent communicator with local Chinese people. These opinions envisaged high Chinese proficiency as an integral part of the participants' ideal L2 selves. Additionally, it was found that the participants' idealized self-images were intensified by their future plans. As a CEO of a small enterprise, Jackson had a strong aspiration to establish business connections with potential Chinese partners, and being fluent in Chinese was a necessary requirement or at least a comparative advantage in his mind.

12. My main goal is to strengthen my ability to engage with Chinese trading partners successfully, reducing my reliance on translators. (Jackson/R/W6)

The actual gap between the participants' L2 self-concepts and their ideal L2 selves stimulated their motivation to improve their spoken Chinese while they were in China. Like Jackson, most participants made significant efforts to promote their pragmatic competence by seeking opportunities and resources both in and outside Chinese classrooms. As Bennett noted:

13. I guess the motivation here, this may lead to practical use of it (Chinese). . . . That's what you always have to do, get into a group of Chinese friends, like a norm for them all to speak Chinese. You can use it every day. (Bennett/Interview after study abroad)

All the participants reported that their Chinese language learning in China was motivated by a desire to interact with Chinese locals, and expressed a willingness to befriend Chinese people on and outside the campus. In the study abroad setting, the majority of the participants acted on their desires and drew on resources to maximize their participation and enhance their pragmatic competence in the local community.

## 5. Discussion

The present study examined the changes in the motivation of a group of New Zealand students in relation to Chinese learning, and the interactions between their shifting motivation and their learning and sociocultural contexts during a six-week period of study abroad in China. Overall, the findings presented above reveal that the participants' Chinese learning motivation was a dynamic process, shaped by the ongoing interaction between their ideal L2 selves, their L2 self-concepts, and the study abroad context. Their learning purposes also had a close connection with motivational shifts. This result is in line with findings from prior research (e.g., [13,15,27]), which has consistently reported that second language learners' motivation was a shifting construct emerging from continuous interactions with study abroad contextual conditions.

Beyond generally demonstrating the dynamic nature of the participants' Chinese learning motivation, however, this research has further illustrated the gap between their L2 self-concepts and their ideal L2 selves, which was elicited by the distance between their perceived learning achievements in New Zealand and their actual learning and usage experiences in China. This gap generally reinforced the internalization of practical learning goals (e.g., moving to China, career development, Chinese fluency) among the participants, and motivated them to make sustainable efforts to reduce their disadvantage in the target language community, experienced in various challenges in academic and social settings. Thus, their Chinese learning motivation showed an instrumental-value inclination during their academic sojourn in China.

Regarding the participants' motivational change during their period of study abroad, this research found that almost all of them reported enhanced motivation after their arrival in China, and that they expected to sustain their motivation to learn and use Chinese in the future, regardless of their original motivation in New Zealand. This is in accordance with Du and Jackson [13]. Thus, short-term study abroad programs need to be encouraged as a central way to sustain and heighten learners' motivation to study Chinese as an additional language. Moreover, study abroad should be considered not only as a contextual factor influencing SL/FL learners' motivation to learn and use the target language, but also as an important component that can trigger learners' self-image construction and transformation [44]. The latter process had a complex association with the participants' motivational change in the present research, and longitudinal research is needed to examine the influence of study abroad experiences on Chinese language learners' motivational sustainability after returning to their home language context.

In addition, it was found that the participants' motivational fluctuations were often closely related to their individual experiences and perceived achievement, especially their pragmatic competence enhancement. Hence, program administrators and teachers need to pay more attention to students' sociocultural backgrounds and different learning needs [45]. At the same time, teachers should consider issues such as pedagogical mode and instructional pace as well as course content, and should provide bespoke courses for students learning Chinese in study abroad contexts. Specifically, program administrators and teachers should provide opportunities and resources for authentic communication to further the students' oral Chinese abilities and build their sense of achievement. However, this finding is different from Allen's [11] research, which indicated that heightened French learning motivation occurred for learners considering study abroad as a language learning experience, but not for participants with pragmatic orientations. A possible explanation behind this might have something to do with different target language proficiency levels of the participants in the two studies. In particular, most of the participants in Allen's study (about 70%) were advanced French learners, whereas almost all the participants (13/15) involved in the present research were beginner and intermediate students. In this regard, future research needs to pay more attention to participants' proficiency level and its impact on SL/FL learning motivation in study abroad contexts. Longitudinal studies involving longer durations of residence in the target language community are necessary in order to explore the interaction between learners' language proficiency improvement and their motivational change.

The finding concerning the interplay between the participants' motivation and learning and their sociocultural contexts is in agreement with studies by Du and Jackson [13] and Gao [14], which both

demonstrated that foreign language learners exhibited a more elaborate learning motivation after relocating to study abroad settings, with the motivation becoming more self-determined. According to Isabelli-García [46], motivation had a reciprocal relationship with participation in social networks; participants showed a strong motivation for engagement into the local community to improve their Chinese proficiency, and, in turn, this engagement promoted their knowledge about Chinese society and oral Chinese used in the daily social milieu. In this sense, classroom learning focusing on linguistic knowledge only serves as one part of language learning experiences during study abroad, and language use outside the classroom should be considered as a critical learning form to maximize the participants' participation in their language practice community abroad. This study showed that seven participants displayed an instrumental-value inclination to learn Chinese, but even though these participants were not in the majority, their motivational change before and after moving to China suggested that learning and sociocultural contexts could have profound influences on foreign language learners' motivation. Hence, it is necessary for researchers to continue investigating the interaction between study abroad students' motivational change and their contextual reality.

## 6. Conclusions

Drawing on Dörnyei's [31] L2 Motivational Self System (L2MSS) and Deci and Ryan's [25] concept of motivation, the present research has examined the evolving Chinese learning motivation of 15 New Zealand students and its interaction with their learning and sociocultural contexts during their academic sojourn in a Chinese university. Analysis of data from reflective journals and two rounds of interviews prior to and after their study abroad trip suggested that most of the participants in this inquiry became more motivated, and expected to make sustainable efforts to learn Chinese after their period of study abroad had ended. At the same time, motivational fluctuations often emerged during their study abroad, influenced by the Chinese pedagogical environment (e.g., instructional style, course content) and personal language learning and use experiences. The students' shifting motivations were promoted by ongoing interactions between their L2 self-concepts, their ideal L2 selves, and the changing learning and sociocultural contexts. These interactions motivated students to use different strategies to make their Chinese learning sustainable, alongside the improvement of their linguistic knowledge and spoken Chinese. Thus, teachers and educational administrators need to consider how to help CAL learners better adapt to the new Chinese language environment and facilitate their learning motivation [47].

The current investigation only involved New Zealand students studying abroad in China, and any generalization of the results to all international students in China or other language learning settings should be undertaken with caution. This research was mainly based on data from reflective journals and interviews. Although the reflection entries were collected weekly and various strategies were used to ensure the trustworthiness of the research findings, what was reported might be different from what was enacted in the actual setting. With this in mind, a mixed-methods approach with both quantitative and qualitative perspectives can be adopted to indicate the dynamics of motivational development of wider learner populations. Additionally, our position as Chinese researchers meant that, to some extent, our data may reflect the predispositions of a Chinese research tradition, especially some aspects with reference to the Chinese pedagogical environment. Despite these limitations, however, we believe that the results of this study provide fresh insights into understanding motivational changes among CAL language learners, and suggest some ways to keep their Chinese learning motivation and practices sustainable. This research also calls for more attention to the role of learning in both formal and informal settings in sustaining international students' language learning and use in the long term [48,49].

**Author Contributions:** Formal analysis, Y.F.G. and M.M.; Funding acquisition, Y.F.G.; Investigation, Y.F.G.; Methodology, Y.F.G.; Project administration, Y.F.G., M.M. and T.P.H.; Resources, M.M.; Writing—original draft, Y.F.G.; Writing—review & editing, Y.F.G., T.P.H., and C.W. All authors have read and agreed to the published version of the manuscript.

**Funding:** This research was funded by University of Macao, Macao SAR, China: SRG2020-00001-FED.

**Acknowledgments:** We would like to thank Xuesong (Andy) Gao and Yawen Han for their support and the reviewers for their professional comments and suggestions.

**Conflicts of Interest:** The authors declare no conflict of interest.

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
