# Peer review of "Sustaining International Students’ Learning of Chinese in China: Shifting Motivations among New Zealand Students during Study Abroad"

_sustainability, doi:10.3390/su12156289_

Round 1
Reviewer 1 Report
The research is qualitative. I feel that a quantitative analysis would enhance the quality of the paper. Dornyei has published some surveys and tests to measure motivation that the author could consider for this paper or further research.
Author Response
Thanks a lot for your time and kind comment. We agree with you that the quantitative design or approach can enhance “rigor and systematicity in data-gathering and analysis as well as comparability and replicability of data, and generalizability to wider populations” (Ushioda & Dörnyei, 2012, p. 401). The present research mainly aimed to examine the change in the motivation of New Zealand students’ Chinese learning before and after their study abroad in China, also we focused on the interaction between their shifting motivation and learning and sociocultural contexts. Thus, we used two rounds of interviews and reflective journals to address the questions. At the same time, using a quantitative perspective, such as survey questionnaires, to compare their motivational change can also be a helpful direction and method, and thus, according to your suggestion, we indicate the significance of quantitative perspective in the conclusion part. Please kindly refer to page 13.
Many thanks again.

Reviewer 2 Report
This study is an inquiry of New Zealand students’ Chinese learning motivations in a typical Study-Abroad context. Interview and reflective diary data were collected in three stages, and a qualitative approach was adopted to analyse the participants’ learning motivation changes. The research was well designed, and the methodology was presented in detail. The research questions are clearly addressed in the findings and discussion sections. Though the whole article is well written, there is still room for improvement before publication.
- Page 3: Two research questions are raised by the authors. But what is the significance of addressing such questions? A few lines about the theoretical and practical significance of the research can be added before moving to the Methodology section.
- Table 2 occupies nearly one full page. However, the contents in the Table are not elaborated at length in the narration. In my opinion, the Table can be deleted.
- As illustrated in 3.2, an interview was conducted among the participants before they started the immersion in China. The purpose of this step was to elicit their pre-immersion learning motivations. However, in Section 4, the motivations were not reported. The heading of 4.1 is “motivational change after arrival in China”, but how about their motivations before arrival in China? One subsection reporting on the pre-immersion motivations is necessary.
- The article does not inform the readers what kind of activities (academic, social, cultural) were arranged during the six-week immersion programme in China. As a result, it is not clear to what extent the students’ motivational change is influenced by the immersion in China. It would be better if this aspect can be briefly discussed.
- Does the research have any implications for CAL teaching and learning? This needs to be highlighted in the Conclusion part.
Author Response
Many thanks for your time and kind comments. We have carefully read and discussed your comments and revised the manuscript according to those. Please kindly refer to our point-to-point responses and relevant revisions in the manuscript.
1.Page 3: Two research questions are raised by the authors. But what is the significance of addressing such questions? A few lines about the theoretical and practical significance of the research can be added before moving to the Methodology section.
Thanks for your suggestion. We agree with you that it is necessary to indicate the theoretical and practical significance of this research, and thus we have added the relevant content according to your suggestion. Please kindly refer to page 2.
2.Table 2 occupies nearly one full page. However, the contents in the Table are not elaborated at length in the narration. In my opinion, the Table can be deleted.
3.As illustrated in 3.2, an interview was conducted among the participants before they started the immersion in China. The purpose of this step was to elicit their pre-immersion learning motivations. However, in Section 4, the motivations were not reported. The heading of 4.1 is “motivational change after arrival in China”, but how about their motivations before arrival in China? One subsection reporting on the pre-immersion motivations is necessary.
Thanks for your comment 2 and 3. In the part 4.1 “Motivational change after arrival in China”, we reported the participants’ motivational enhancement from New Zealand to China and fluctuations during their learning in China. In this part, Table 2 was used to indicate their shifting motivation in details. At the same time, we also expected to show their motivational change in a clear way for readers and thus adopted this table. Through this table, the differences of the participants’ Chinese learning motivation between prior to and after their academic sojourn can be seen. Thus, we would like to keep this table for its value. Thanks for your kind understanding.
4.The article does not inform the readers what kind of activities (academic, social, cultural) were arranged during the six-week immersion programme in China. As a result, it is not clear to what extent the students’ motivational change is influenced by the immersion in China. It would be better if this aspect can be briefly discussed.
Thanks for your comment. This short-term immersion program mainly focused on New Zealand students’ Chinese proficiency improvement, and thus courses of listening, reading, and Chinese character writing accounted for a main proportion. At the same time, other activities such as field trips to local traditional venues were also included in this course to increase their understanding of Chinese local lives, society and culture. These parts have been added and indicated for more details. Please kindly refer to page 4.
5.Does the research have any implications for CAL teaching and learning? This needs to be highlighted in the Conclusion part.
Thanks for your kind comment. This research aimed to further our understanding of Chinese learning motivation in particular and SL/FL motivation in general. We also expect to provide implications for educational stakeholders to facilitate CAL learners’ motivation. According to your suggestion, we have highlighted this and added relevant content in the conclusion part. Please kindly refer to page 13.
Many thanks again.
